# Functional and Transcriptomic Characterization of Postnatal Maturation of ENS and SIP Syncytium in Mice Colon

**DOI:** 10.3390/biom13121688

**Published:** 2023-11-23

**Authors:** Zhihao Wu, Qianqian Wang, Fan Yang, Jiaxuan Wang, Yuying Zhao, Brian A. Perrino, Jie Chen

**Affiliations:** 1Department of General Surgery, Shanghai Children’s Medical Center, Shanghai Jiao Tong School of Medicine, Shanghai 200127, China; 2Division of Gastroenterology and Hepatology, Shanghai Institute of Digestive Disease, Renji Hospital Affiliated to Shanghai Jiao Tong University School of Medicine, Shanghai 200001, China; 3Bio-X Institutes, Key Laboratory for the Genetics of Developmental and Neuropsychiatric Disorders, Ministry of Education, Shanghai Jiao Tong University, Shanghai 200240, China; 4Department of Physiology and Cell Biology, School of Medicine, University of Nevada, Reno, NV 89557, USA

**Keywords:** enteric nervous system, cholinergic neurons, nitrergic neurons, purinergic receptors, SIP syncytium, colonic motility

## Abstract

The interplay of the enteric nervous system (ENS) and SIP syncytium (smooth muscle cells–interstitial cells of Cajal–PDGFRα+ cells) plays an important role in the regulation of gastrointestinal (GI) motility. This study aimed to investigate the dynamic regulatory mechanisms of the ENS-SIP system on colon motility during postnatal development. Colonic samples of postnatal 1-week-old (PW1), 3-week-old (PW3), and 5-week-old (PW5) mice were characterized by RNA sequencing, qPCR, Western blotting, isometric force recordings (IFR), and colonic motor complex (CMC) force measurements. Our study showed that the transcriptional expression of *Pdgfrα*, *c-Kit*, *P2ry1*, *Nos1*, and *Slc18a3*, and the protein expression of nNOS, c-Kit, and ANO1 significantly increased with age from PW1 to PW5. In PW1 and PW3 mice, colonic migrating movement was not fully developed. In PW5 mice, rhythmic CMCs were recorded, similar to the CMC pattern described previously in adult mice. The inhibition of nNOS revealed excitatory and non-propulsive responses which are normally suppressed due to ongoing nitrergic inhibition. During postnatal development, molecular data demonstrated the establishment and expansion of ICC and PDGFRα+ cells, along with nitrergic and cholinergic nerves and purinergic receptors. Our findings are important for understanding the role of the SIP syncytium in generating and establishing CMCs in postnatal, developing murine colons.

## 1. Introduction

The enteric nervous system (ENS) is composed of complex networks of neurons and ganglia, mainly containing excitatory motor neurons (cholinergic responses) and inhibitory motor neurons (nitrergic and purinergic responses) that regulate colonic movement [1,2,3]. The ENS originates from enteric neural crest stem cells (ENCCs), mainly from vagal neural crest cells, as reported previously [4,5,6]. It is well known that the regulation of the excitability of smooth muscle cells (SMC) by the ENS mainly depends on two types of interstitial cells, namely interstitial cells of Cajal (ICCs) and platelet-derived growth factor alpha positive cells (PDGFRα+ cells). ICCs, PDGFRα+ cells, and SMCs are electrically coupled to each other and constitute a functional unit known as the SIP syncytium (SMC/ICC/PDGFRα+ cell) [7,8,9]. Colonic movement is controlled by the coordination between the ENS and SIP syncytium [10]. Nitric oxide (NO) is the dominant inhibitory neurotransmitter responsible for smooth muscle relaxation via ICCs [11,12]. As an excitatory neurotransmitter, acetylcholine (Ach) also induces smooth muscle contraction by binding to M3 receptors on ICCs [13]. In addition, purines bind to purinergic P2Y receptors on PDGFRα+ cells, which are coupled to Gq/11-IP3 pathways, which, in turn, release Ca^2+^ from the endoplasmic reticulum. The increase in intracellular Ca^2+^ activates small conductance calcium-activated potassium channels (SK) which are involved in the relaxation of colonic smooth muscles via membrane hyperpolarization via the SIP syncytium electrical coupling [14].

The ENS-SIP plays an important role in gastrointestinal (GI) motility. The establishment of the ENS is gradually achieved by the proliferation and migration of ENCCs throughout the gut [15,16,17]. During embryonic week (EW) 12, the fetal colon exhibits a dense distribution of the neural network with the expression of excitatory cholinergic neural transmitters, while the expression of inhibitory nitric neural transmitters appears around EW14. Electrical stimulation fails to evoke activity in the ENS of EW 12 or EW 14 [18]. Furthermore, neuronal nitric oxide synthase (nNOS)-expressing nitrergic neurons appear at approximately embryonic day 12.5, while choline acetyltransferase (ChAT)-expressing cholinergic neurons emerge at embryonic day 14.5 [19,20,21]. These studies indicate that the ENCCs present extensive proliferation and migration at various embryonic stages, during which diverse neuron subtypes are expressed, and as a result, immature neuronal electrical activity emerges.

Mice are born with functionally immature GI motility [22]. The development of the ENS in 1-week-old mice is incomplete, based on dendritic morphologies, axon projections, and electrophysiological properties [23,24]. The motility patterns of the early postnatal period (up to 3 weeks after birth) are significantly different from the postweaning period (after 3 weeks) due to changes in diet and the microbiota [25,26,27,28]. At around 3 weeks, mice start weaning and undergo significant changes in diet (solid food), behavior, and physiology, with further development of the ENS toward functional maturation [29].

Despite some important advances in elucidating the mechanisms of neuronal regulation of colonic motility, most experiments still focus on ENS changes during embryonic and postnatal days [30,31], and the complete temporal and spatial postnatal development and malfunction of the ENS and SIP syncytium in the colon have not been fully characterized. Many colonic motor disorders occur during childhood, including Hirschsprung disease, slow transit constipation, and partial colonic obstruction (PCO), but little is known about the normal patterns of development of colonic motility after birth. Therefore, in this study, we selected mice at 1 week postnatal (infancy), 3 weeks postnatal (weaning), and 5 weeks postnatal (juvenile) to simulate the developmental process [32] and to explore the postnatal colonic development of the SIP syncytium during ENS expansion. A recent study revealed the developmental process of ENS in different spatial parts of embryonic and juvenile mice from a morphological and structural perspective, but they have not yet characterized the molecular changes in the ENS during this period [33]. Thus, we first performed molecular analyses of the ENS-SIP syncytium transcript and protein expression changes during the dynamic development of the murine colon. In the mouse colon, a rhythmic neurogenic motor pattern is reliably recorded from the isolated whole mouse colon, called the colonic motor complex (CMC) [34,35,36]. So, we next performed functional analyses including isometric force recording and measurements of CMC to investigate the functional changes during the dynamic development of the juvenile murine colon.

## 2. Materials and Methods

### 2.1. Experimental Animals

We used two animal facilities. C57BL/6 mice were purchased from The Jackson Laboratory (Bar Harbor, ME, USA), generated in-house at the University of Nevada, Reno, for Wes (simple western system) experiments (see below, Wes section). All other experiments were performed at Shanghai Jiao Tong University School of Medicine. C57BL/6 mice were obtained from the Experimental Animal Center of Shanghai Jiao Tong University School of Medicine. The experiments were performed according to the rules of the Institutional Animal Use and Care Committee at the University of Nevada, Reno, and in compliance with the rules of the Guide for the Care and Use of Laboratory Animals of the Science and Technology Commission of China (STCC Publication No. 2, revised 1988). The protocol was approved by the Committee on the Ethics of Animal Experiments of Shanghai Jiao Tong University School of Medicine (Permit Number: Hu 686-2009). The mice were fed ad libitum with unrestricted access to water. The mice were maintained at 23 °C under a 12 h light/dark cycle. All of the mice used in this study were age-matched postnatal 1-week-old (PW1), 3-week-old (PW3), and 5-week-old (PW5).

### 2.2. Total RNA Isolation and Quantitative Real-Time Polymerase Chain Reaction (qRT-PCR)

Mice were sacrificed with an overdose of isoflurane inhalation followed by cervical dislocation. The entire colons were quickly removed before stripping the mucosal layers via sharp dissection. When the whole colon was opened to expose the mucosa along the mesenteric border, the proximal colon displayed a characteristic thick mucosal fold. This thick mucosal fold section was used for proximal preparation (1 or 2 cm below the ileocecal sphincter). For the distal colon, we used muscle preparations from 2 cm above the internal anal sphincter. These preparations were subsequently rinsed with Krebs–Ringer bicarbonate (Krebs) solution before being frozen in liquid nitrogen. Total RNA was isolated from proximal and distal colon tissues using the Direct-zol RNA Purification Kit (Zymo Research, Irvine, CA, USA) following which, the quality of total RNAs was determined using NanoDrop 2000 (Thermo Scientific, Waltham, MA, USA). qRT-PCR was performed using Fast SYBR Green chemistry (Applied Biosystems, Foster City, CA, USA) on the QuantStudio3 Real-Time PCR System (Applied Biosystems). Regression analysis of the mean values of technical triplicate qPCRs for the log10-diluted cDNA was used to generate standard curves. Unknown amounts of messenger RNA (mRNA) were plotted relative to the standard curve for each set of primers and graphically plotted using Microsoft Excel. This gave the transcriptional quantification of each gene relative to the endogenous glyceraldehyde 3-phosphate dehydrogenase (*Gapdh*) (NM_008084), the standard after log transformation of the corresponding raw data. The primers used for quantifying the target transcripts are listed in Appendix A.

### 2.3. RNA Sequencing and Data Analyses

Samples with an RNA integrity number (RIN) exceeding 8.5 were processed for preparation of libraries using a NEBNext Ultra II RNA Library Prep Kit from Illumina (Cat#E7770). RNA sequencing was performed using an Illumina NovaSeq 6000 platform at the DNA facility of Novogene Co., Ltd. (Beijing, China). Deep sequencing was conducted with 150 bp paired-end reads. Each biological replicate was considered as a separate sample. The raw reads were preprocessed to remove low-quality reads and adapters using Trimmomatic (a flexible read trimming tool for Illumina NGS data). All reads were then mapped to the genome (reference: Ensembl Mus musculus GRCm38.p6) using the HISAT2 method. The transcript was then assembled and quantified using the String Tie method. The expression matrix was based on FPKM and subject to Log2 (FPKM+1) normalization.

### 2.4. Wes Simple Western Automated Capillary Electrophoresis and Immunodetection

Muscles were snap-frozen in liquid N2 and stored at −80 °C. For analysis, frozen samples were placed into 0.5 mL ice-cold lysis buffer, homogenized using a bullet blender (1 stainless steel bead, speed 6), and centrifuged at 16,000× *g* (4 °C, 10 min), and the supernatants were stored at −80 °C [37,38]. Protein concentrations were determined by Bradford assay, using bovine γ-globulin as the standard [37]. Analysis of protein expression was performed according to the Wes User Guide using a ProteinSimple Wes instrument (www.proteinsimple.com, accessed from 1 January 2019 to 1 January 2020) [37,38]. Each sample was mixed with fluorescent 5× Master Mix, incubated at 95 °C for 5 min, and then loaded into a Wes 12–230 kDa prefilled plate, along with a biotinylated protein ladder, blocking buffer, primary antibodies, ProteinSimple HRP-conjugated anti-rabbit secondary antibody, luminol peroxide, and washing buffer. The plates and capillary cartridges were placed into the Wes for electrophoresis and chemiluminescence immunodetection by a CCD camera using default settings. Compass software for SW4.1.0 (ProteinSimple) was used to acquire and analyze the data and generate gel images and chemiluminescence intensities. Protein expression levels are expressed as the chemiluminescence intensity area under the primary antibody peak per µg protein. All primary and secondary antibodies used are described in Appendix A.

### 2.5. Western Blot

For the detection of protein abundance, we also employed the conventional protein assay (Western blot). Briefly, 10 µL colonic tissue lysates were separated using electrophoresis, transferred to a nitrocellulose membrane, and detected by the relevant antibodies (described in Appendix A) following the manufacturer’s instructions.

### 2.6. Isometric Force Recording (IFR)

Mice were sacrificed as described above. Proximal and distal colon segments were obtained as described above. Muscle strips of length 8 mm and width 2 mm were cut parallel to the circular muscle layer from both proximal and distal colon segments. A silk thread (USP 5/0) was attached to both ends of the muscle strips, and the strips were immersed in warm (37 °C) oxygenated (95% O_2_ and 5% CO_2_) Krebs solution in 25 mL organ baths. Mechanical activity was recorded on a computer connected to an isometric force transducer (Axon Instruments, Foster City, CA, USA). The contraction of the colonic smooth muscle strips was displayed using the software LabChart version 7.0. The colons of PW1 mice were too fragile to record contractions. The area under the curve (AUC) for 5-min recordings of spontaneous contractions was measured. The AUC was normalized to the control to compare the effect of the drug. We tested the effects of MRS2500 and LNNA (Tocris Bioscience, Ellisville, MO, USA), and atropine (Sigma-Aldrich, St. Louis, MO, USA) on contractions. The information of all drugs used are shown in Appendix A.

### 2.7. Colonic Motor Complexes

The mice were sacrificed as described above. The entire colon was harvested and dissected to remove the mesentery and fat tissues. The feces were carefully expelled with a repeated injection of Krebs solution. A glass capillary tube was inserted carefully through the empty lumen and fixed to the floor of the dish. The colon segment was rinsed with warm Krebs solution (36.5 °C, 5% CO_2_, and 95% O_2_) and stabilized for 30–40 min to recover the colonic contraction activity. A silk thread (USP 5/0) was applied to the proximal, mid, and distal colon to connect the force transducer. CMC activity was recorded using an isometric force transducer (Masterflex, Chicago, IL, USA). We also tested the effects of MRS2500, LNNA, and atropine on CMCs at PW1, PW3, and PW5 colons. The change in frequency and amplitude was analyzed for 20-min recordings before and after each drug application.

### 2.8. Statistical Analysis (Clarify Transcriptional, Protein, IFC, and CMC Experiments)

Quantitative analysis of transcripts was expressed as mean ± SD, and all other data were expressed as means ± SEM. “n” represents the number of animals. All statistical analyses were performed using GraphPad Prism. Paired and unpaired Student’s *t* tests were used to compare groups of data, and differences were defined as significant at *p* < 0.05.

## 3. Results

### 3.1. Characterization of the Transcriptome in Young Murine Colon

To understand the development of the myenteric ENS in the young murine colon, RNA sequencing was performed on colonic tunica muscularis during the early postnatal period; PW1, PW3, and PW5. Compared to PW1, 1016 differentially expressed genes (DEGs) (log2 (fold change, FC) ≥ 1 or ≤−1, *p* < 0.05) were detected in PW3 mice, of which 401 and 615 genes were significantly upregulated and downregulated, respectively (Appendix A). Also, 1235 DEGs, including 463 upregulated and 772 downregulated genes, were screened in PW5 mice (Appendix A). The results of the gene ontology analyses revealed that the majority of enriched DEGs were involved in cell division, cell growth, and cell differentiation (Appendix A), suggesting that young mice were at a stage of rapid growth and development. Expectedly, some representative genes that participated in the functioning of the myenteric ENS exhibited temporal and spatial expression diversity during this period (Figure 1A), such as well-known genes involved in neuronal and glial cell subtypes (*Nos1* and *Gfap*), synapsins (*Snap25*, *Syn1*, *Syn2*, and *Syt1*), semaphorins (*Sema3a*, *Sema3b*, *Sema3c*, and *Sema3f*), voltage-gated sodium channels (*Scn7a*, *Scn1b*, and *Scn3b*), and voltage-gated potassium channels (*Kcnq2*, *Kcnq4*, and *Kcnq5*). The results of qRT-PCR further confirmed that the level of transcripts of *Kcna1* and *Scn7a*, as well as *Gfap*, was markedly increased in the colon of PW3 and PW5 mice (Figure 1B–D). Although the transcriptional level of *Scn5a*, *Snap25*, *Syn2*, and *Syt1* showed no significant increase in the colon of PW3 and PW5 mice, there was still a slight upward trend (Appendix A) compared to PW1 colons. In protein analysis, GFAP expression was increased in the proximal and distal colon with development (Figure 1E). In order to further detect the ENS development extent, we performed qRT-PCR on represented markers of ENS progenitors (*Sox10*, *Ret*, and *Phox2b*) and other neuronal and glial cell subtypes (*Chat* and *S100b*). The transcriptional level of *Ret* exhibited a significant decreased tendency in the colon of PW3 and PW5 mice compared with PW1 mice. The level of transcripts of *Sox10*, *Phox2b*, *Chat*, and *S100b* showed no statistical differences, but *S100b* presented an increasing trend. These findings suggest that some neurons and glial cells undergo increasing phenotypic differentiation during early colonic development.

### 3.2. Expression of Key Genes and Proteins in SIP Syncytium following Postnatal Development

We focused on the gene expression pattern in ICCs and PDGFRα+ cells, two types of interstitial cells which are electrically coupled to SMCs and function as a SIP syncytium in controlling GI motility [39]. The qRT-PCR and Wes analyses revealed that *Pdgfrα* transcript and PDGFRα protein levels were gradually increased in both the proximal and distal colon in PW3 and PW5 mice (Figure 2A,B). The protein level of SK3 (KCNN3), which encodes Ca^2+^-activated K channels and is highly expressed in colonic PDGFRα+ cells, showed a decreased expression in the proximal colons of PW3 and an increased expression in the proximal and distal colon of PW5 (Figure 2C).The mRNA levels of *c-Ki*t, the signature gene of ICC, were also markedly elevated in the proximal colon of PW3 and PW5 mice (Figure 2D,E) but the mRNA of *c-Kit* exhibited no changes in the expression in the distal colon up to PW5 (Figure 2D). The protein levels of c-Kit also showed a significant increasing trend in PW5 mice. The protein expression of ANO1, known to be the main pacemaker conductance in ICCs [40], was also significantly increased in PW5 mice (Figure 2F). These results indicate that the development of interstitial cells, including ICCs and PDGFRα+ cells, gradually progressed in both the proximal and distal colons after birth.

### 3.3. Expression of Excitatory and Inhibitory Motor Neurons following Postnatal Development

We also investigated the genes involved in inhibitory and excitatory motor responses. The results of qRT-PCR confirmed that the transcriptional expression of *Nos1* was dramatically increased in both the proximal and distal colon of PW3 and PW5 mice (Figure 3A). Interestingly, *Nos1* expression was more robust in the proximal colon than in the distal colon (Figure 3A). Protein data show that NOS1 protein was abundantly expressed in the proximal colon of PW3 and PW5 mice (Figure 3B). In contrast, transcriptional expression of the purinergic receptor P2Y (*P2ry1*), the marker gene of PDGFRα+ cells, was increased in the distal colon of PW3 and PW5 but showed no significant change in the proximal colon (Figure 3C). The mRNA level of *P2ry1* was much higher in the distal colon than the proximal colon (Figure 3C). However the abundance of P2RY1 protein was gradually elevated in the distal colon tissues, but the elevated tendency was also observed in the proximal colon of PW3 and PW5 mice (Figure 3D). Unlike the expression pattern of P2ry1, the transcriptional level of Solute carrier family 18 member A3 (*Slc18a3*, vesicular acetylcholine transporter, VAChT), the signature gene of cholinergic neurons, was significantly increased in the proximal colon of PW3 and PW5 mice (Figure 3E) and in the distal colon of PW5 mice. *Slc18a3* mRNA level was higher in the proximal colon than in the distal colon (Figure 3E). But the protein expression of VAChT, encoded by the Slc18a3 gene, was also significantly increased in the distal colon of PW3 and PW5 mice compared to that in the proximal colon (Figure 3F). Similarly, nNOS+ expression increased in both the proximal and distal colons, whereas purinergic receptor (P2ry1) expression gradually increased in the distal colon rather than the proximal colon during the postnatal period. Taken together, these data demonstrate that nNOS and VAChT in the proximal colons and P2Y1 receptors in the distal colons might have dominant functional roles during postnatal development.

### 3.4. Excitatory and Inhibitory Motor Responses of Colonic Contractions

#### 3.4.1. Responses of Colonic Contractions in the Proximal Colon

To further investigate the functional role of inhibitory and excitatory neurons during different stages of development and growth, IFR assays with pharmacological approaches were performed on colon smooth muscle strips from PW3 and PW5 mice. The colons of PW1 mice were too fragile to be used for contractile measurements. To demonstrate the effects of purinergic responses on the spontaneous contractions, we employed the P2Y1 receptor antagonist MRS2500 (1 µM) to block the effect of purinergic responses. MRS2500 had minimal effect on proximal colon smooth muscle spontaneous contractions (area under the curve, AUC) in both PW3 (106.9 ± 0.08%, *n* = 6) and PW5 (115.6 ± 0.05%, *n* = 6) mice compared to the control (before MRS2500 treatment, *n* = 6, Figure 4A–E). Subsequent application of LNNA (100 µM), a blocker of nNOS, dramatically increased the AUC and tone in proximal colon muscle strips from PW3 (125.4 ± 0.12%, *p* < 0.001) and PW5 (247.3 ± 0.43%, *p* < 0.001) mice compared to MRS2500 (Figure 4A–E). These data suggest that nitrergic inhibitory responses in the proximal colon are present at PW3 and PW5. Continuous application of atropine (1 µM, a muscarinic receptor antagonist) in the presence of MRS2500 and LNNA decreased LNNA responses to 103.4 ± 0.08% (*p* < 0.001) in PW3 and 118.3 ± 0.16% (*p* < 0.001) in PW5 mice compared to LNNA responses (Figure 4A–E). These data suggest that both the cholinergic excitatory and nitrergic inhibitory responses are fully developed in the proximal colon at PW5. In contrast, up to PW5, purinergic inhibition of spontaneous contractions was negligible in the proximal colon. These data also demonstrate that the cholinergic excitatory and nitrergic inhibitory responses were not fully developed in the proximal colon of PW3 mice compared to PW5 mice.

#### 3.4.2. Responses of Colonic Contractions in the Distal Colon

In smooth muscle strips from the distal colon, MRS2500 (1 µM) had no obvious effects on the spontaneous contractions from PW3 mice (*n* = 6, Figure 5A,B,E) but significantly increased the AUC to 124.1 ± 0.10% in PW5 mice (*p* < 0.05, Figure 5C–E). Subsequent treatment with LNNA (100 µM) did not show a further increase in the AUC of the distal colon in PW3 (107.1 ± 0.05%) and PW5 (113.2 ± 0.06%) compared to those of MRS2500 treatment (Figure 5E). Atropine significantly inhibited the effect of MRS2500 and LNNA responses in the distal colon in PW3 mice (*p* < 0.05) and in PW5 mice (*p* < 0.01) (Figure 5E).

### 3.5. Impact of Neuron Regulators on the CMCs

To further investigate the fundamental mechanism of colonic motility regulated by ENS, CMC assays were performed on the entire colon of PW1, PW3, and PW5 mice with the application of MRS2500, LNNA, and atropine. In PW1 mice, the proximal colon showed a high frequency of spontaneous contraction without propulsive migratory movement, and the middle and distal colon displayed CMC activities (Figure 6A). However, PW3 and PW5 mice generated CMC activities (Figure 6B,C). The CMC patterns displayed higher amplitudes and frequencies in the proximal, middle, and distal colons of PW3 and PW5 mice compared to PW1 mice (Figure 6D,E). These data suggest that CMCs developed higher amplitudes and lower frequencies as a function of age.

#### 3.5.1. Responses of CMCs with the Application of LNNA

The effects of LNNA on the CMCs were also investigated. LNNA (100 µM) had no significant effect on the frequency and amplitude of the non-propulsive spontaneous contractions of the PW1 proximal colon (Figure 7D,E). However, LNNA induced an increase in the frequency of spontaneous contractions from the middle to distal colon of PW1 mice (Figure 7A). It seemed that LNNA might only induce non-propulsive contractions in the middle and distal colon due to a lack of a migrating complex in the proximal colon of PW1 mice. At PW3, the CMC amplitudes were not affected by LNNA (Figure 7B,E). However, LNNA showed a significant effect on CMC frequencies of the proximal colon but had no effect on the middle and distal colon of PW3 mice (Figure 7B,D). PW5 colons displayed a normal CMC pattern (Figure 7C). LNNA induced a high frequency of spontaneous contractions in the proximal colon and abolished the CMC propagation toward the middle and distal colon (Figure 7C–E). These data suggest that nNOS expression appeared at PW3 in the proximal colon and modulated excitatory responses due to ongoing nitrergic inhibition during the generation of CMC.

#### 3.5.2. Responses of CMCs with the Application of Atropine

We also tested the development of cholinergic neurons, which play an excitatory role on colonic motility through ICCs. Treatment with atropine (1 µM) had no significant effect on the frequency and amplitude of CMCs in PW1 middle and distal colons (Figure 8A,D,E). PW3 mice showed different responses to atropine from the proximal to the distal colon (Figure 8B,D,E). However, atropine significantly repressed the frequency and amplitude of CMCs in PW5 mice (Figure 8C–E).

#### 3.5.3. Responses of CMCs with the Application of MRS2500

Moreover, purinergic blocker MRS2500 (1.0 µM) had no significant effect on CMCs in PW1 and PW3 mice (Appendix A). However, MRS2500 decreased the frequency and amplitude of CMC in the distal colon of PW5 mice (Appendix A). This suggests that purinergic regulation became more dominant on distal CMC as the mice aged from PW1 to PW5.

## 4. Discussion

This research focused on the development of colonic motility associated with the expression of neuronal and SIP syncytium markers and ENS function, as well as the roles of NO, purines, and Ach on the spontaneous contractions and CMCs during the postnatal period.

The current study shows that enteric neurons undergo expansion and function during the early colon developmental process, and the development of interstitial cells, including ICCs and PDGFRα+ cells, gradually progresses in both the proximal and distal colon after birth. The transcriptional expression of Pdgfrα, c-Kit, P2ry1, Nos1, and Slc18a3 and the protein expression of nNOS, c-Kit, and ANO1 significantly increase with age. The responses of spontaneous contractions to inhibitory and excitatory stimuli were not fully developed in the proximal colon up to PW3 mice. At PW5, normal patterns of CMCs were generated, displaying high amplitudes and lower frequencies, similar to murine adult CMCs, and the inhibition of nNOS revealed dominant excitatory and non-propulsive responses due to ongoing nitrergic inhibition during the generation of CMC.

The ENS consists of a complex network of neurons and ganglia and shares a close connection with interstitial cells which are spread throughout the GI wall [41,42,43]. During ENS development, several ENCCs proliferate and migrate extensively along the murine gut [15,16,17]. Despite significant findings in ENS development during the embryonic stage, evidence regarding the development of neuronal subtypes or coordinated electrical activity in the ENS is scarce and the associated effects on colonic motility at a young age are unknown. The development and establishment of coordinated electrical activity are crucial for normal colonic motility and transit function [18]. In the current study, gene ontology analyses revealed that the majority of enriched DEGs were involved in cell division, cell growth, and cell differentiation. The maturation and functionality of the myenteric ENS exhibited temporal and spatial diversity of gene expression during this period. The levels of transcripts and proteins which are involved in neuronal and glial development were markedly increased in the colon of PW3 and PW5 mice. These findings suggest that neurons and glial cells become increasingly functional during the early colonic development.

Purinergic inhibition can occur due to electric coupling via gap junctions between SMC and PDGFRα+ cells. Kurahashi et al. reported that PDGFRα+ cells exist in the GI tract of mice [8]. The activation of purinergic receptors, mainly P2Y1, induces hyperpolarization due to the activation of SK channels in PDGFRα+ cells, resulting in smooth muscle relaxation [44]. In this study, *Pdgfra* transcript levels were gradually increased in both the proximal and distal colon in PW3 and PW5 mice. However, expression of P2ry1, the marker gene of PDGFRα+ cells, was increased in the distal colon of PW3 and PW5 mice but showed no significant change in the proximal colon. The mRNA levels of *P2ry1* were also much higher in the distal colon than the proximal colon of PW3 and PW5 mice. However, the abundance of the P2RY1 protein was gradually elevated in both proximal and distal colon tissues. No obvious differences in the expression of SK3 were observed between the proximal and distal colon of PW3 and PW5 mice, but it still had a slightly increasing trend. In functional experiments, MRS2500, a specific P2Y1 receptor antagonist, abolished CMCs in the distal colon of PW5 mice. Similarly, isometric muscle contraction experiments indicated that MRS2500 increased the AUC of contractions. In contrast, MRS2500 did not significantly affect the CMCs or smooth muscle contractions of distal colons in PW1 and PW3 mice, indicating that purinergic receptors were not fully developed at those stages compared to PW5. These results indicate that purinergic receptors are not functionally developed until the mice are 5 weeks of age. The inhibitory function of purinergic receptors was essential to maintain the rhythm of propulsive colonic motility [45,46,47].

NO is also involved in the inhibitory regulation of smooth muscle contraction and the generation of CMCs through ICCs [30,48,49,50,51]. Nitrergic neurons are responsible for sustained relaxations which are required for processes such as propulsion and storage activity [47]. A previous study revealed that the inhibitory regulation of nitrergic neurotransmitters was vital to maintain regular colonic motility and colonic transmission [52]. Thus, we hypothesized that the development of nNOS neurons and ICC should involve the generation of regular rhythmicity including contractions and CMCs. The transcriptional and protein levels of c-Kit and ANO1, the signature genes of ICCs, were markedly elevated in the proximal colon during postnatal development but displayed no distinct changes in the distal colon, suggesting that the proximal colon may be subject to a dominant nitrergic inhibitory input during postnatal development. Functional data demonstrated that LNNA dramatically increased the AUC with tone development in the proximal colon but not in the distal colon at PW3. In the CMC experiments, the proximal colon did not show detectable propulsive CMC, and LNNA had no effect on the frequencies and amplitudes at PW1. PW3 and PW5 proximal colons generated CMCs and showed a significant negative effect of LNNA on the frequency of CMCs. These data suggest that nNOS expression appeared at PW3 in the proximal colon, and the inhibition of nNOS showed dominant excitatory and non-propulsive responses due to ongoing nitrergic inhibition during the generation of CMC.

Cholinergic neurons release the neuromuscular transmitter Ach, which couples to muscarinic receptors (mainly M3) and regulates the pacemaker activity via activation of the ANO1 channel in ICCs [53,54,55]. CMC generation has been shown to be due to coordinated firing of large populations of excitatory and inhibitory neurons at ~2 Hz [56] but may involve ICC for neurotransmission. In this study, we found that the transcriptional and protein level of VAChT was significantly increased in the proximal and distal co of PW5 mice. The transcriptional level of *Slc18a3* was significantly increased in the proximal colon of PW3 and PW5 mice and in the distal colon of PW5 mice. The *Slc18a3* mRNA level was higher in the proximal colon than in the distal colon. These data suggest that cholinergic neurons developed rapidly in the proximal colon but relatively slowly in the distal colon during the postnatal period. Previous reports have demonstrated that cholinergic neurons were more widespread in the proximal murine colon compared to the distal colon [57,58]. Atropine suppressed LNNA-augmented contractions in PW5 colonic muscle strips and significantly suppressed the frequency and amplitude of CMC in PW5 colons but had no effect on PW1 colons, suggesting that functional cholinergic neurotransmission in the proximal colon was not developed by PW1 but became fully developed and subject to regulation at PW5.

We have previously reported functional studies of the SIP syncytium in adult mice (postnatal 10 weeks) [52]. These results showed that ICCs were predominant in the proximal colon of adult mice, while the distal colon had a more characteristic distribution of PDGFRα+ cells. Due to the action of ENS-NO/ICC and purine/PDGFRα+ cells in the colon, a pressure gradient from the proximal colon to the distal colon is formed. Compared to adult mice, the findings in this report of the CMC motility patterns and responses to drugs in early adolescent mice (PW5) showed similar trends.

The pathophysiology of the most common congenital gut motility defects is likely to be multifactorial. The development of ENS activity in the human gut influences not only neural development per se but also impacts the associated cell types in the neuromuscular and SIP syncytium which ultimately dictate motility. Our previous studies in PCO mice showed that colonic dysmotility disorders can also be affected by the developmental status of ICC, PDGFRα+ cells, and purinergic, nitrergic, and cholinergic neurons and their corresponding receptors [59]. However, the development and maturation of these factors in the control of colonic motility in young children have not been thoroughly investigated; therefore, the current study may help to guide the development of new therapeutic options for congenital gut motility defects.

During postnatal development of the proximal, middle, and distal colon, the molecular expression data demonstrate the temporal and spatial maturation of ICC and PDGFRα+ cells, as well as the nitrergic and cholinergic nerves and purinergic receptors. PW5 colons generated a normal or adult pattern of CMCs with proper pharmacological responses. Cholinergic and nitrergic responses in the proximal colon and purinergic responses in the distal colon showed dominant functional roles with the increase in age. Our findings are important in understanding the roles of the ENS and the SIP syncytium and their specific receptors in the generation of mature CMCs in young murine colons. The unique process of development of the SIP syncytium and enteric motor neurons in young mice may provide some understanding of the mechanisms and facilitate the treatment of pediatric patients with GI motility disorders.

## Figures and Tables

**Figure 1 biomolecules-13-01688-f001:**
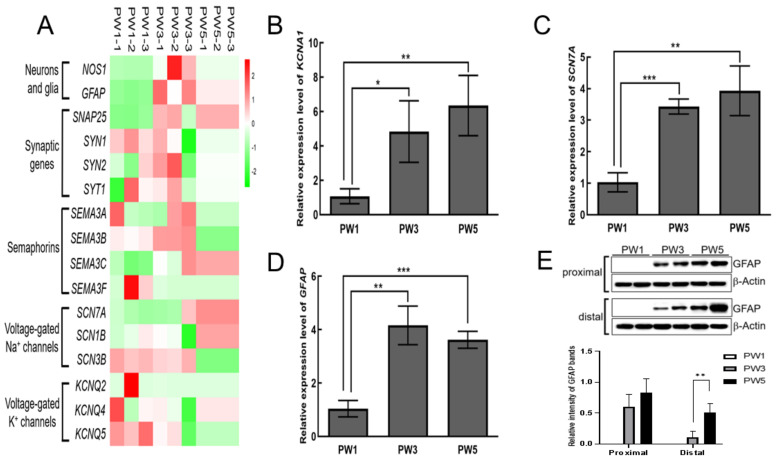
Differential expression of key genes involved in the developing enteric neuron system. (**A**): Heat plot showing the expression of representative genes during postnatal murine colon development in each biological replicate of PW1, PW3, and PW5 (top) based on RNA sequencing results with lenient analysis parameters. Significant mRNA changes between low relative expression (green) and high relative expression (red) represent false discovery rate values ≤ 0.05. (**B**–**D**): Relative quantitative mRNA levels of *Kcna1*, *Scn7a*, and *Gfap* in the whole colon of PW1, PW3, and PW5 samples. The bar graphs shown in (**B**–**D**) represent mean ± SD (*n* = 3, “*n*” means the number of animals). (**E**): The result of immunoblotting of GFAP protein in the proximal colon and distal colon of the three samples. *Gapdh* and *β–Actin* were used as reference in qRT–PCR and immunoblotting assays, respectively. * *p* < 0.05; ** *p* < 0.01, *** *p* < 0.001. Original images of (**E**) can be found in Appendix A.

**Figure 2 biomolecules-13-01688-f002:**
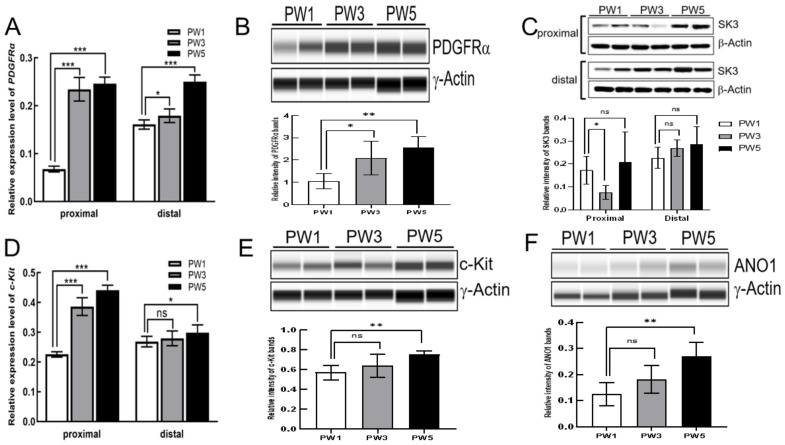
The expression of representative interstitial cell genes. (**A**): The result of quantitative real-time PCR of *Pdgfra* mRNA levels in the proximal colon (*n* = 4) and the distal colon (*n* = 4) of PW1, PW3, and PW5 samples. (**B**): Wes analysis of PDGFRα protein in the proximal colon of the three samples. (**C**): The result of immunoblotting of SK3 protein in the proximal colon and the distal colon of the three samples. (**D**): The result of quantitative real-time PCR of *c-Kit* mRNA levels in the proximal colon (*n* = 4) and the distal colon (*n* = 4). (**E**,**F**): Wes analysis of c-Kit and ANO1 proteins in the proximal colon (*n* = 4). The bar graphs shown in (**A**,**B**), and (**D**–**F**) represent mean ± SD (*n* = 4). “*n*” means the number of animals. *Gapdh* and *b-ACTIN* or *g-ACTIN* were used as reference in qRT-PCR and immunoblotting assays, respectively. Samples were compared using unpaired *t*-test. * *p* < 0.05; ** *p* < 0.01; *** *p* < 0.001; ns, no significant difference. Original images of (**B**,**C**,**E**,**F**) can be found in Appendix A.

**Figure 3 biomolecules-13-01688-f003:**
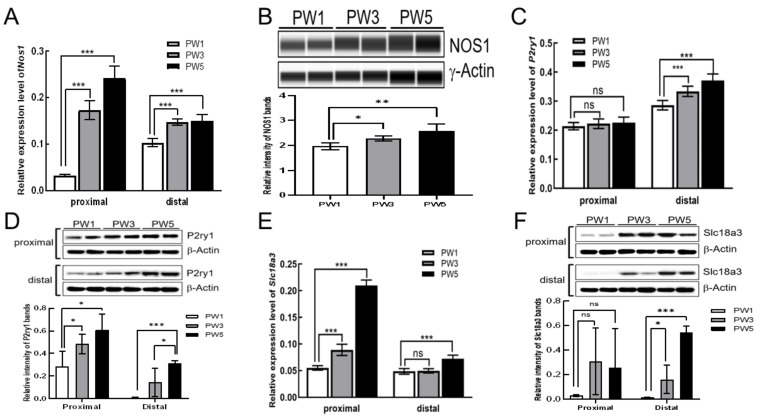
The expression of representative genes in inhibitory neurons and excitatory neurons. (**A**,**C**,**E**): The result of quantitative real-time PCR of *Nos1*, *P2ry1*, and *Slc18a3* mRNA levels in PW1, PW3, and PW5 of the proximal colon (*n* = 4) and the distal colon (*n* = 4) of each period. (**B**): Wes analysis of NOS1 protein in the proximal colon of the three samples. (**D**,**F**): The result of immunoblotting of P2ry1 and Slc18a3 proteins in the proximal colon and the distal colon of the three samples. The bar graphs shown in (**A**–**C**,**E**) represent mean ± SD (*n* = 4). “*n*” means the number of animals. Gapdh and β-ACTIN or γ-ACTIN were used as reference in qRT-PCR and immunoblotting assays, respectively. Samples were compared using unpaired *t*-test. * *p* < 0.05; ** *p* < 0.01; *** *p* < 0.001; ns, no significant difference. Original images of (**B**,**D**,**F**) can be found in Appendix A.

**Figure 4 biomolecules-13-01688-f004:**
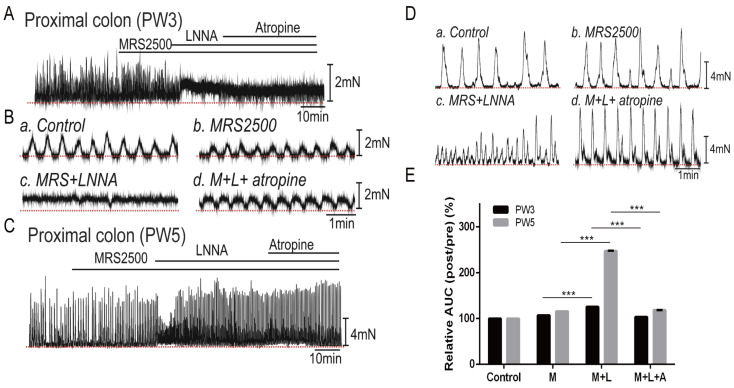
The effects of neuronal antagonists on spontaneous contraction of smooth muscle in the proximal colon. (**A**): Combined treatment with MRS2500 (1.0 µM), LNNA (100.0 µM), and atropine (1.0 µM) on the isolated murine proximal colon of PW3. (**B**): The smooth muscle contraction of PW3 proximal colon without treatment (**a**), or upon the application of MRS2500 (**b**), LNNA (**c**), and atropine (**d**). (**C**): Continuous treatment of MRS2500, LNNA, and atropine on the isolated proximal colon of PW5 murine. (**D**): The smooth muscle contraction of PW5 proximal colon without treatment (**a**), or upon the application of MRS2500 (**b**), LNNA (**c**), and atropine (**d**). (**E**): The contractile activity of proximal colon smooth muscle measured by the ratio of AUC within 5 min before and after (post/pre) MRS2500 (M), LNNA (L), and atropine (A) treatment (*n* = 6). “*n*” means the number of animals. The bar graphs represent mean ± SE. *** *p* < 0.001. The red dotted line in the figure is the baseline.

**Figure 5 biomolecules-13-01688-f005:**
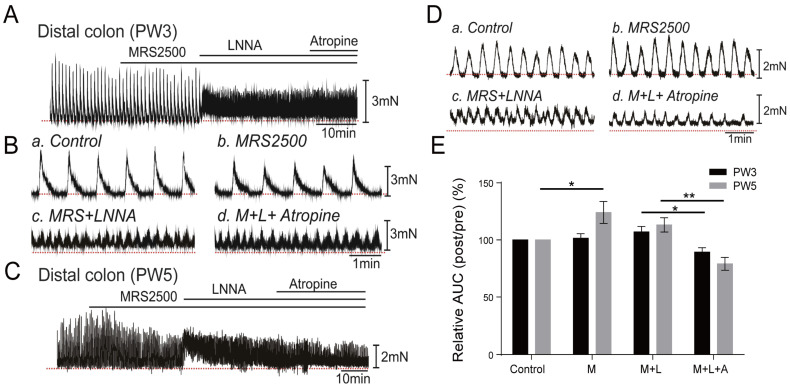
The effects of neuronal antagonists on spontaneous contraction of smooth muscle in the distal colon. (**A**): Combined treatment of MRS2500 (1.0 µM), LNNA (100.0 µM), and atropine (1.0 µM) on the isolated distal colon of PW3 murine. (**B**): The smooth muscle contraction of PW3 distal colon without treatment (**a**), or upon the application of MRS2500 (**b**), LNNA (**c**), and atropine (**d**). (**C**): Combined treatment of MRS2500, LNNA, and atropine on the isolated distal colon of PW5 murine. (**D**): The smooth muscle contraction of PW5 distal colon without treatment (**a**), or upon the application of MRS2500 (**b**), LNNA (**c**), and atropine (**d**). (**E**): The contractile activity of distal colon smooth muscle measured by the ratio of AUC within 5 min before and after (post/pre) MRS2500 (M), LNNA (L), and atropine (A) treatment. The bar graphs represent mean ± SE (*n* = 6). “n” means the number of animals. * *p* < 0.05; ** *p* < 0.01. The red dotted line in the figure is the baseline.

**Figure 6 biomolecules-13-01688-f006:**
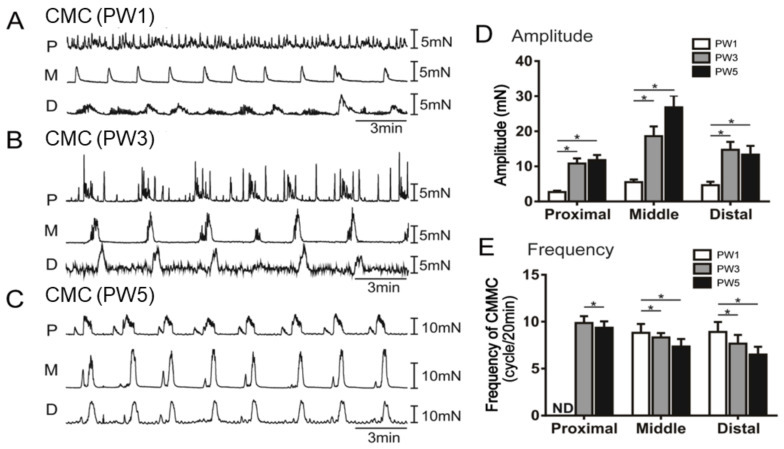
The colonic motor complexes of various colon sections in young mice. (**A**–**C**): The contraction features of isolated proximal colon (P), middle colon (M), and distal colon (D) under normal conditions in PW1 (**A**), PW3 (**B**), and PW5 (**C**) mice. (**D**): The quantified CMC frequency of each colon segment in PW1, PW3, and PW5 mice. (**E**): The quantified CMC amplitude of each colon segment in PW1, PW3, and PW5 mice. The bar graphs shown in panels (**D**,**E**) represent mean ± SE (*n* = 15). “*n*” means the number of animals. * *p* < 0.05.

**Figure 7 biomolecules-13-01688-f007:**
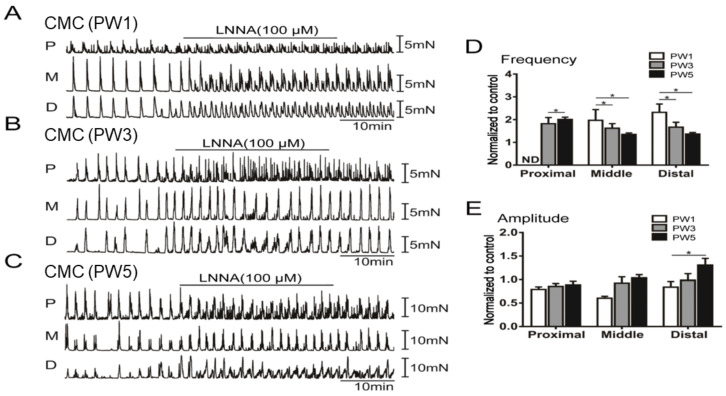
The excitatory effects of the nitrergic neuron blocker LNNA on CMC in the young murine colon. (**A**–**C**): The contraction features of isolated proximal (P), middle (M), and distal colon (D) treated with LNNA (100.0 µM) from PW1 (**A**), PW3 (**B**), and PW5 mice (**C**). (**D**): The CMC frequency of each colon section treated with LNNA from PW1, PW3, and PW5 mice (*n* = 6). ND represent not determined. (**E**): The CMC amplitude of each colon section treated with 100.0 µM LNNA from PW1, PW3, and PW5 mice (*n* = 6). “*n*” means the number of animals. The bar graphs shown in panel D and E represent mean ± SE. * *p* < 0.05.

**Figure 8 biomolecules-13-01688-f008:**
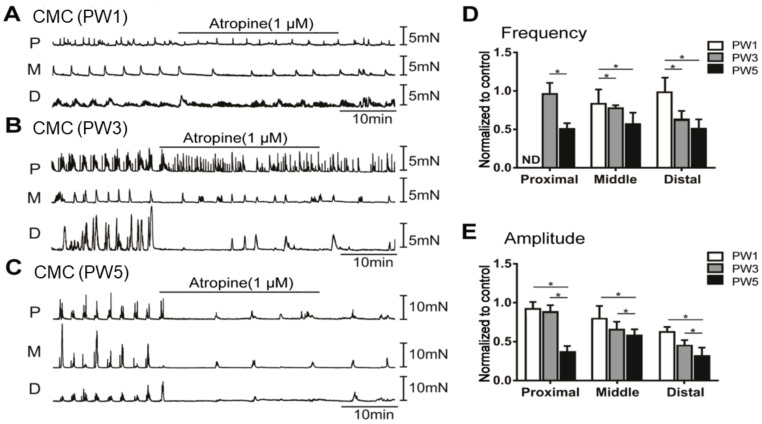
The inhibitory effects of the muscarinic receptor antagonist atropine on CMC of the young murine colon. (**A**–**C**): Contractile responses of isolated proximal colon (P), middle colon (M), and distal colon (D) treated with atropine (1.0 µM) from PW1 (Panel **A**), PW3 (Panel **B**), and PW5 (Panel **C**) mice. (**D**): The CMC frequency of each colon segment treated with atropine from PW1, PW3, and PW5 mice. ND represents not determined. (**E**): The CMC amplitude of each colon segment treated with atropine from PW1, PW3, and PW5 mice. The bar graphs shown in panels (**D**,**E**) represent mean ± SE (*n* = 5). “*n*” means the number of animals. * *p* < 0.05.

## Data Availability

The datasets generated for this study can be found in the Gene Expression Omnibus (GEO) database (accession number GSE186568).

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
