# Peer review of "Functional and Transcriptomic Characterization of Postnatal Maturation of ENS and SIP Syncytium in Mice Colon"

_biomolecules, 2023, doi:10.3390/biom13121688_

Round 1
Reviewer 1 Report
Comments and Suggestions for Authors
This is a descriptive study evaluating postnatal development and function of the ENS and SIP syncytium in mouse colon. The authors demonstrate temporal and spatial differences in the mRNA and protein expression of genes related to the ENS and SIP syncytium in proximal and distal mouse colon from postnatal week 1 to 5. They additionally evaluate ENS and SIP syncytium dependent functions including contractility and the colonic migrating motor complex (CMMC). Major findings include: 1. increasing transcriptional expression of PDGFRa, cKIT, NOS1, and Slc18a3, and increasing protein expression of GFAP, nNOS, and cKIT with age, 2. cholinergic excitatory and nitrergic inhibitory responses were not fully mature in both proximal and distal mice colon until PW5, 3. CMMCs exhibited higher amplitudes and lower frequencies as a function of age, and 4. purinergic regulation became more mature on distal CMMC when mice aged.
This study is significant as the temporal and spatial postnatal development of the ENS and SIP syncytium in the colon has not been fully characterized. This study is not particularly innovative, it uses a standard approach to quantify mRNA/protein levels and measure contractility; the study is mostly descriptive, thus no novel concepts are introduced. The approach is fairly adequate and appropriate. However, where I have a major concern is the definition of "maturity" or "maturation".
1. The authors mention both terms multiple times throughout the manuscript; however, they are never defined. The authors insinuate that increased transcriptional or protein expression of a gene equals maturity; however, I'm not sure that is true. Maturity of a tissue or cell type may require specific transcriptional and protein expression changes temporally and spatially. For example, during ENS development, ENS progenitors capable of becoming enteric neurons (neurogenesis) or enteric glial cells (gliogenesis) express SOX10, RET, and PHOX2B. However, neurogenic commitment requires downregulating SOX10 and maintaining RET expression, while gliogenesis requires maintaining SOX10 and downregulating RET. PHOX2B expression is maintained in all enteric neurons and only enteric glia. Enteric neuronal subtypes then express a variety of markers such as ChAT, nNOS, 5-HT, etc. The full extent of enteric glial diversity remains unknown, but many glia will express GFAP and/or S100B. The authors conclude in line 212-213 that "neurons and glial cells became increasingly mature" but this is based on the increasing expression levels of only NOS1 and GFAP. It may be more appropriate to show the expression of progenitor markers SOX10, RET, PHOX2B, followed by EN or EGC markers ChAT, nNOS, 5-HT, GFAP, S100B over time, to demonstrate "maturation". Furthermore, increasing expression levels of NOS1 and GFAP may signal that there are a higher number of enteric neurons and glia, rather than increasing "maturity" of a finite number of cells.
The authors also mention in lines 420-423 "levels of transcripts and proteins which are involved in neuronal and glial development were markedly increased...suggest that neurons and glial cells become increasingly mature". Again, simply increased expression doesn't equal maturity, and the classic ENS developmental genes mentioned above were not described in this paper.
2. Same comments for Figure 2 and maturity/maturation. What determines maturity of the SIP syncytium? I'm not sure that increasing expression of PDGFRa and cKIT equals "maturity" of PDGFRa cells and ICCs. A higher expression level of those proteins may simply signify increased expression of a finite number of cells, or it may signify the same level of expression in an increasing number of cells. Either way, I don't think increased expression signifies "maturity", that would have to be determined by either morphology or better yet, function of the cells.
3. The authors mention that the CMMCs generated in PW5 mouse colon were similar to murine adult CMMCs but that data is never shown. It would be really helpful to show the waveforms for murine adult CMMCs for comparison/control so we can see that they are similar.
4. Finally, the authors mention that two animal facilities were used (U of Nevada in Reno and Shanghai Jiao Tong U School of Medicine. It is well known that microbiome and diet can influence ENS development and function. I'm assuming that the microbiome and diets were likely different at these two locations. Thus, if the transcriptional data was determined from mice in Shanghai and the protein data was determined from mice in Reno, is it possible that the differences seen between transcriptional and protein data are due to microbiome and/or diet? Please comment.
I enjoyed reviewing your manuscript and look forward to your comments and revision.
Comments on the Quality of English Language
Line 25 rhythum should be rhythm
Author Response
Thank you very much for taking the time to review this manuscript. We have made modifications and supplements to the article according to your feedback, and have marked the modified areas in red, hoping to meet your requirements.
|
Comments 1: The authors mention both terms multiple times throughout the manuscript; however, they are never defined. The authors insinuate that increased transcriptional or protein expression of a gene equals maturity; however, I'm not sure that is true. Maturity of a tissue or cell type may require specific transcriptional and protein expression changes temporally and spatially. For example, during ENS development, ENS progenitors capable of becoming enteric neurons (neurogenesis) or enteric glial cells (gliogenesis) express SOX10, RET, and PHOX2B. However, neurogenic commitment requires downregulating SOX10 and maintaining RET expression, while gliogenesis requires maintaining SOX10 and downregulating RET. PHOX2B expression is maintained in all enteric neurons and only enteric glia. Enteric neuronal subtypes then express a variety of markers such as ChAT, nNOS, 5-HT, etc. The full extent of enteric glial diversity remains unknown, but many glia will express GFAP and/or S100B. The authors conclude in line 212-213 that "neurons and glial cells became increasingly mature" but this is based on the increasing expression levels of only NOS1 and GFAP. It may be more appropriate to show the expression of progenitor markers SOX10, RET, PHOX2B, followed by EN or EGC markers ChAT, nNOS, 5-HT, GFAP, S100B over time, to demonstrate "maturation". Furthermore, increasing expression levels of NOS1 and GFAP may signal that there are a higher number of enteric neurons and glia, rather than increasing "maturity" of a finite number of cells.
The authors also mention in lines 420-423 "levels of transcripts and proteins which are involved in neuronal and glial development were markedly increased...suggest that neurons and glial cells become increasingly mature". Again, simply increased expression doesn't equal maturity, and the classic ENS developmental genes mentioned above were not described in this paper.
|
|
Response 1: Thank you for pointing this out. We totally agree with this comment. Therefore, we have removed the concept of maturity in the manuscript and added the RT-PCR results of Sox10, Ret, Phox2b, Chat, and S100b in the manuscript. The Sox10, Phox2b, and Chat did not show a significant trend of change in PW3 and PW5 mice compared to PW1 mice. However, the Ret showed a decreased trend in PW3 and PW5 mice compared to PW1 mice. Although the mRNA of S100b did not show statistical significance, it showed a gradually increasing trend. And the results of Ret and S100b were highly confirmed to the results of GFAP exhibited in Fig 1D and E, which might mean the beginning of gliogenesis progress in PW3 mice. The results were shown in the supplementary Figure 3 (E-I) and the descriptive text were in lines 222-228.
|
|
Comments 2: Same comments for Figure 2 and maturity/maturation. What determines maturity of the SIP syncytium? I'm not sure that increasing expression of PDGFRa and cKIT equals "maturity" of PDGFRa cells and ICCs. A higher expression level of those proteins may simply signify increased expression of a finite number of cells, or it may signify the same level of expression in an increasing number of cells. Either way, I don't think increased expression signifies "maturity", that would have to be determined by either morphology or better yet, function of the cells. |
|
Response 2: Thank you so much for your advice. We have changed the concept of maturity with the “development” or “increasing number of cells” in the manuscript. |
|
Comments 3: The authors mention that the CMMCs generated in PW5 mouse colon were similar to murine adult CMMCs but that data is never shown. It would be really helpful to show the waveforms for murine adult CMMCs for comparison/control so we can see that they are similar. |
|
Response 3: Thank you so much for your advice. This manuscript mainly displays the ENS and SIP syncytial changes in mice before adulthood, and we have reported functional studies of the SIP syncytium in adult mice (postnatal 10 wks) in our previous article. (https://pubmed.ncbi.nlm.nih.gov/30510374/ Lu, C., Huang, X., Lu, H.L., Liu, S.H., Zang, J.Y., Li, Y.J., Chen, J.and Xu, W.X. Different distributions of interstitial cells of Cajal and platelet-derived growth factor receptor-α positive cells in colonic smooth muscle cell/interstitial cell of Cajal/platelet-derived growth factor receptor-α positive cell syncytium in mice. World J Gastroenterol. 2018, 24, 4989-5004). |
|
Comments 4: Finally, the authors mention that two animal facilities were used (U of Nevada in Reno and Shanghai Jiao Tong U School of Medicine. It is well known that microbiome and diet can influence ENS development and function. I'm assuming that the microbiome and diets were likely different at these two locations. Thus, if the transcriptional data was determined from mice in Shanghai and the protein data was determined from mice in Reno, is it possible that the differences seen between transcriptional and protein data are due to microbiome and/or diet? Please comment. |
|
Response 4: Thank you so much for your advice. Our mouse strains were all C57 BL/6, which were stable species and commonly used as blank controls in many studies. All mice were purchased from The Jackson Laboratory (Bar Harbor, ME, USA) and all were kept in a barrier environment that complied with SPF standards. The drinking water, feed, daily routine, and environment were controlled the same, but there may still be some impacts. This is one limitation of our experiment, and we will conduct further research on this influencing factor in the future. |
|
Response to Comments on the Quality of English Language |
|
Point 1: Line 25 rhythum should be rhythm |
|
Response 1: Thank you so much for your advice. We have revised this mistake. |

Reviewer 2 Report
Comments and Suggestions for Authors
In the present study, the authors characterized the postnatal mouse colon by RNA sequencing, qPCR, western blotting, isometric force recordings (IFR) and colonic migrating motor complex (CMMC) force measurements, and described molecular and functional development of the enteric nervous system (ENS) and SIP syncytium (smooth muscle cells–intestinal cells of Cajal–PDGFRα+ cells). The present study provides interesting and important data to understand the role of the ENS and SIP syncytium for the generation of mature gastrointestinal motility in the postnatal mouse colon. One of the interesting results is molecular and functional difference between the proximal and distal colons. Overall, the manuscript is well written. Nonetheless, I raise the following points to be revised in the present manuscript.
Major:
Although the authors stated that no obvious difference in P2RY1 protein abundance was observed in the proximal colon of PW3 and PW5 mice (Fig. 3D), it is unclear whether the P2RY1 band intensities are different between PW1, PW3 and PW5 or not. It appears that P2RY1 band intensities of PW1 may be lower than those of PW3 and PW5. Therefore, the authors should clarify this issue by quantifying the P2RY1 band intensities in Fig. 3D.
Minor:
In sections 3.4 and 3.5, I suggest that the authors divide each section into subsections to clearly present each data. For example, "3.4.1. Responses in the proximal colon", "3.4.2. Responses in the distal colon" and so on.
Comments on the Quality of English Language
Minor editing of English language is required.
Author Response
Thank you very much for taking the time to review this manuscript. We have made modifications and supplements to the article according to your feedback, and have marked the modified areas in red, hoping to meet your requirements.
|
Point-by-point response to Comments and Suggestions for Authors |
|
Comments 1: Although the authors stated that no obvious difference in P2RY1 protein abundance was observed in the proximal colon of PW3 and PW5 mice (Fig. 3D), it is unclear whether the P2RY1 band intensities are different between PW1, PW3 and PW5 or not. It appears that P2RY1 band intensities of PW1 may be lower than those of PW3 and PW5. Therefore, the authors should clarify this issue by quantifying the P2RY1 band intensities in Fig. 3D.
|
|
Response 1: Thank you so much for your advice. We have added bar charts in Figures 1E, 2C, 3D, and 3F. And by quantifying the P2RY1 bands intensities in Fig.3D, the tendency is significantly elevated in PW3 and PW5 mice. So, we have revised the description of the figures in the original text.
|
|
Comments 2: In sections 3.4 and 3.5, I suggest that the authors divide each section into subsections to clearly present each data. For example, "3.4.1. Responses in the proximal colon", "3.4.2. Responses in the distal colon" and so on. |
|
Response 2: Thank you so much for your advice. We have added titles to each subsection of sections 3.4 and 3.5 to present the content of each section more clearly. |
|
Response to Comments on the Quality of English Language |
|
Point 1: Minor editing of English language is required. |
|
Response 1: Thank you so much for your advice. We have made modifications to some inappropriate language in the manuscript. |

Reviewer 3 Report
Comments and Suggestions for Authors
Wu et al., in the paper entitled " Functional and Transcriptomic Characterization of Postnatal Maturation of ENS and SIP syncytium in Mice Colon", investigated the role of ENS and SIP syncytium in the development of colonic migration motor complex and colon function in the developing mice. Besides, they characterize the transcriptome of mice colon and validate them at three different age points.
On the other hand, the authors look at the maturation of complex colonic function in the developing colon.
It is a well-designed work, with a clear objective and an adequate methodology, although some aspects require more detail. The results of this study are interesting although they differ from published work.
From my point of view, the work is interesting, but the method of the study and the conclusions pointed out by the authors in the work would improve if some experiments were completed, some aspects were clarified and the discussion was deepened in aspects related to more published work.
Suggestions from my review are described below and significant revision of this manuscript would be required before the publication in the journal.
1. Authors used pharmacological blockers while recording colonic contraction to conclude that cholinergic excitatory and nitrergic inhibitory responses are developed in the developing mice. They need to use electrical field stimulation (EFS) to recapitulate how excitatory and inhibitory neurotransmitters released from these neurons (due to EFS) and bind with the respective receptors located on the components of SIP syncytium and subsequent change in the smooth muscle contraction. Secondly the result doesn’t match with the representative trace, e.g. (i) in Fig 4Dd, after atropine application the contraction should be inhibited but the raw trace still showed high magnitude spike (ii) 5Bc, after LNNA, contraction should increase (as shown in 5E) but raw trace does not show that)
2. Authors need to use reference to show recording mechanical forces from three locations of colon can be represented as CMMCs as video recording of colonic motility and converting them to spatiotemporal map is widely accepted method for CMMCs readout (e.g. Roberts et al., 2007).
3. The author mentioned “the transcriptional level of Scn5a, Snap25, Syn2, and Syt1 209
also showed various degrees of increase in PW3 and PW5 colons (Fig. S3 A-D) compared
to PW1 colon”. Is this increase significant as I cannot see any significance mark in the Fig. S3 A-D. Heatlpot (Fig 1A) doesn’t correspond to your statement “…….Syt1 also showed various degrees of increase in PW3 and PW5 colons…..(Line 209, P-5)”
4. Author mentioned in Page 6, Line 233: “In contrast, the mRNA and protein levels of c-Kit, the signature gene of ICC, were also markedly elevated in the proximal colon of PW3 and PW5 mice (Fig. 2D & E)”. However, protein levels of c-Kit in the proximal colon of PW3 is not significant (Fig. 2E). Same is true for ANO1 (Fig. 2F).
5. It would be nice to put a bar graph in Fig 2 C, 3D and 3F
6. Roberts et al. showed that CMMC developed in postnatal D10 mice, authors need to justify why they found different result.
7. Application of LNNA should increase both frequency and amplitude of contraction as per Barnes and spencer, 2015. Figure 7C shows overall decrease in amplitude.
Minor:
Figure S3. Legend missing “mice” after PW1, PW3, and PW5.
Figure: 2D: Y axis – c-kit, “c” is missing
Page 6, Line 233: “in contrast” doesn’t make sense.
Fig 3A. Y axis- NOS1 woul be Nos1
Reference:
Curr Biol. 2022 Oct 24;32(20):4483-4492.e5. doi: 10.1016/j.cub.2022.08.030. Epub 2022 Sep 6.
Regional cytoarchitecture of the adult and developing mouse enteric nervous system
Ryan Hamnett 1, Lori B Dershowitz 1, Vandana Sampathkumar 2, Ziyue Wang 3, Julieta Gomez-Frittelli 4, Vincent De Andrade 5, Narayanan Kasthuri 2, Shaul Druckmann 6, Julia A Kaltschmidt 7
Am J Physiol Gastrointest Liver Physiol 2007 Mar;292(3):G930-8. doi: 10.1152/ajpgi.00444.2006.
Development of colonic motility in the neonatal mouse-studies using spatiotemporal maps
Rachael R Roberts 1, Jessica F Murphy, Heather M Young, Joel C Bornstein
Clin Exp Pharmacol Physiol. 2015 May;42(5):485-95. doi: 10.1111/1440-1681.12380.
Can colonic migrating motor complexes occur in mice lacking the endothelin-3 gene?
Kyra J Barnes 1, Nick J Spencer
Comments on the Quality of English Language
Wu et al., in the paper entitled " Functional and Transcriptomic Characterization of Postnatal Maturation of ENS and SIP syncytium in Mice Colon", investigated the role of ENS and SIP syncytium in the development of colonic migration motor complex and colon function in the developing mice. Besides, they characterize the transcriptome of mice colon and validate them at three different age points.
On the other hand, the authors look at the maturation of complex colonic function in the developing colon.
It is a well-designed work, with a clear objective and an adequate methodology, although some aspects require more detail. The results of this study are interesting although they differ from published work.
From my point of view, the work is interesting, but the method of the study and the conclusions pointed out by the authors in the work would improve if some experiments were completed, some aspects were clarified and the discussion was deepened in aspects related to more published work.
Suggestions from my review are described below and significant revision of this manuscript would be required before the publication in the journal.
1. Authors used pharmacological blockers while recording colonic contraction to conclude that cholinergic excitatory and nitrergic inhibitory responses are developed in the developing mice. They need to use electrical field stimulation (EFS) to recapitulate how excitatory and inhibitory neurotransmitters released from these neurons (due to EFS) and bind with the respective receptors located on the components of SIP syncytium and subsequent change in the smooth muscle contraction. Secondly the result doesn’t match with the representative trace, e.g. (i) in Fig 4Dd, after atropine application the contraction should be inhibited but the raw trace still showed high magnitude spike (ii) 5Bc, after LNNA, contraction should increase (as shown in 5E) but raw trace does not show that)
2. Authors need to use reference to show recording mechanical forces from three locations of colon can be represented as CMMCs as video recording of colonic motility and converting them to spatiotemporal map is widely accepted method for CMMCs readout (e.g. Roberts et al., 2007).
3. The author mentioned “the transcriptional level of Scn5a, Snap25, Syn2, and Syt1 209
also showed various degrees of increase in PW3 and PW5 colons (Fig. S3 A-D) compared
to PW1 colon”. Is this increase significant as I cannot see any significance mark in the Fig. S3 A-D. Heatlpot (Fig 1A) doesn’t correspond to your statement “…….Syt1 also showed various degrees of increase in PW3 and PW5 colons…..(Line 209, P-5)”
4. Author mentioned in Page 6, Line 233: “In contrast, the mRNA and protein levels of c-Kit, the signature gene of ICC, were also markedly elevated in the proximal colon of PW3 and PW5 mice (Fig. 2D & E)”. However, protein levels of c-Kit in the proximal colon of PW3 is not significant (Fig. 2E). Same is true for ANO1 (Fig. 2F).
5. It would be nice to put a bar graph in Fig 2 C, 3D and 3F
6. Roberts et al. showed that CMMC developed in postnatal D10 mice, authors need to justify why they found different result.
7. Application of LNNA should increase both frequency and amplitude of contraction as per Barnes and spencer, 2015. Figure 7C shows overall decrease in amplitude.
Minor:
Figure S3. Legend missing “mice” after PW1, PW3, and PW5.
Figure: 2D: Y axis – c-kit, “c” is missing
Page 6, Line 233: “in contrast” doesn’t make sense.
Fig 3A. Y axis- NOS1 woul be Nos1
Reference:
Curr Biol. 2022 Oct 24;32(20):4483-4492.e5. doi: 10.1016/j.cub.2022.08.030. Epub 2022 Sep 6.
Regional cytoarchitecture of the adult and developing mouse enteric nervous system
Ryan Hamnett 1, Lori B Dershowitz 1, Vandana Sampathkumar 2, Ziyue Wang 3, Julieta Gomez-Frittelli 4, Vincent De Andrade 5, Narayanan Kasthuri 2, Shaul Druckmann 6, Julia A Kaltschmidt 7
Am J Physiol Gastrointest Liver Physiol 2007 Mar;292(3):G930-8. doi: 10.1152/ajpgi.00444.2006.
Development of colonic motility in the neonatal mouse-studies using spatiotemporal maps
Rachael R Roberts 1, Jessica F Murphy, Heather M Young, Joel C Bornstein
Clin Exp Pharmacol Physiol. 2015 May;42(5):485-95. doi: 10.1111/1440-1681.12380.
Can colonic migrating motor complexes occur in mice lacking the endothelin-3 gene?
Kyra J Barnes 1, Nick J Spencer
Author Response
Thank you very much for taking the time to review this manuscript. We have made modifications and supplements to the article according to your feedback, and have marked the modified areas in red, hoping to meet your requirements.
|
Point-by-point response to Comments and Suggestions for Authors |
|
Comments 1: Authors used pharmacological blockers while recording colonic contraction to conclude that cholinergic excitatory and nitrergic inhibitory responses are developed in the developing mice. They need to use electrical field stimulation (EFS) to recapitulate how excitatory and inhibitory neurotransmitters released from these neurons (due to EFS) and bind with the respective receptors located on the components of SIP syncytium and subsequent change in the smooth muscle contraction. Secondly the result doesn’t match with the representative trace, e.g. (i) in Fig 4Dd, after atropine application the contraction should be inhibited but the raw trace still showed high magnitude spike (ii) 5Bc, after LNNA, contraction should increase (as shown in 5E) but raw trace does not show that)
|
|
Response 1: Thank you very much for your suggestion. Firstly, we admitted the shortcoming of the research that we didn’t use EFS in spontaneous contractions. Indeed, we tried EFS in several PW3 mice, however it’s so fragile, EFS could nearly abolish the spontaneous contractions, and affect the results of drug application. We used AUC (area under curve) to recapitulate contractions of smooth muscle. The trace showed high magnitude after atropine application, while the contraction baseline and frequency were decreased, therefore AUC was lowered in Fig. 4Dd and Fig. 5Bc. Similarly, although the high magnitude was lowered after LNNA application, the contraction baseline and frequency were lifted, thus, AUC was increased.
|
|
Comments 2: Authors need to use reference to show recording mechanical forces from three locations of colon can be represented as CMMCs as video recording of colonic motility and converting them to spatiotemporal map is widely accepted method for CMMCs readout (e.g. Roberts et al., 2007). |
|
Response 2: Thank you very much for your advice. We agreed that video recording was specific and representative for CMCs, also, the traces of CMCs for data analysis was also frequently used in similar research (e.g. Koh SD, et al, Proc Natl Acad Sci U S A. 2022 May 3;119(18): e2123020119; Wang Q, et al, J Neurogastroenterol Motil. 2019 Oct 30;25(4):589-601. ). |
|
Comments 3: The author mentioned “the transcriptional level of Scn5a, Snap25, Syn2, and Syt1 209 also showed various degrees of increase in PW3 and PW5 colons (Fig. S3 A-D) compared to PW1 colon”. Is this increase significant as I cannot see any significance mark in the Fig. S3 A-D. Heatlpot (Fig 1A) doesn’t correspond to your statement “…….Syt1 also showed various degrees of increase in PW3 and PW5 colons…..(Line 209, P-5)” |
|
Response 3: Thank you very much for your advice. We expanded the samples and retested the four indicators of Scan5a, Snap25, Syn2, and Sym1, and found that compared to PW1, the P values of these four indicators in the colon of PW3 and PW5 were not statistically significant. In order to highlight the reliability of the results, we chose a box plot to display this part of the results in Fig. S3. We have annotated the statistical identification in the bar chart, and changed the description of the result to “a slight upward trend”. |
|
Comments 4: Author mentioned in Page 6, Line 233: “In contrast, the mRNA and protein levels of c-Kit, the signature gene of ICC, were also markedly elevated in the proximal colon of PW3 and PW5 mice (Fig. 2D & E)”. However, protein levels of c-Kit in the proximal colon of PW3 is not significant (Fig. 2E). Same is true for ANO1 (Fig. 2F). |
|
Response 4: Thank you very much for your advice. We have made adjustments to the description of the results of this figure. |
|
Comments 5: It would be nice to put a bar graph in Fig 2 C, 3D and 3F |
|
Response 5: Thank you so much for your advice. We have added bar charts in Figures 1E, 2C, 3D, and 3F. |
|
Comments 6: Roberts et al. showed that CMMC developed in postnatal D10 mice, authors need to justify why they found different result. |
|
Response 6: Thank you very much for your comments and advice. Roberts et al in 2007 described that the CMMC (including interval, duration, velocity, and length propagated) in P10 mice were calculated through the construction of multiple spontaneous contractions. However, CMCs were not merely construction of spontaneous contractions. It is a highly developed motor complex. It was a motor complex generated from the proximal to the distal colon, which included the rhythm, pattern, magnitude and frequency of the whole colon. We made comparison from different perspectives, and may draw different conclusion and results. |
|
Comments 7: Application of LNNA should increase both frequency and amplitude of contraction as per Barnes and spencer, 2015. Figure 7C shows overall decrease in amplitude. |
|
Response 7: Thank you very much for your advice and comments. Based on our results, LNNA increased the CMCs frequency in PW1, PW3, and PW5 mice, while it did not increase the amplitudes especially in PW1 and PW3 mice. It was also an interesting and challenging question to us. The possible reason could be that CMCs were not fully developed in PW3 and PW1 mice. We would focus on the development in our further studies. |
|
Response to Comments on the Quality of English Language |
|
Point 1: Figure S3. Legend missing “mice” after PW1, PW3, and PW5.ine 25 rhythum should be rhythm |
|
Response 1: Thank you so much for your advice. We have revised this mistake. |
|
Point 2: Figure: 2D: Y axis – c-kit, “c” is missing |
|
Response 2: Thank you so much for your advice. We have corrected the error in this image. |
|
Point 3: Page 6, Line 233: “in contrast” doesn’t make sense. |
|
Response 3: Thank you so much for your advice. We have corrected the error in the manuscript. |
|
Point 4: Fig 3A. Y axis- NOS1 woul be Nos1 |
|
Response 4: Thank you so much for your advice. We have corrected the error in this image. |
|
5. Response to Comments on the Reference |
|
Point 1: Curr Biol. 2022 Oct 24;32(20):4483-4492.e5. doi: 10.1016/j.cub.2022.08.030. Epub 2022 Sep 6. Regional cytoarchitecture of the adult and developing mouse enteric nervous system Ryan Hamnett 1, Lori B Dershowitz 1, Vandana Sampathkumar 2, Ziyue Wang 3, Julieta Gomez-Frittelli 4, Vincent De Andrade 5, Narayanan Kasthuri 2, Shaul Druckmann 6, Julia A Kaltschmidt 7 |
|
Response 1: Thank you so much for your advice. We have added this reference in our manuscript, which ranked 33. |
|
Point 2: Am J Physiol Gastrointest Liver Physiol 2007 Mar;292(3):G930-8. doi: 10.1152/ajpgi.00444.2006. Development of colonic motility in the neonatal mouse-studies using spatiotemporal maps Rachael R Roberts 1, Jessica F Murphy, Heather M Young, Joel C Bornstein |
|
Response 2: Thank you so much for your advice. We have added this reference in our manuscript, which ranked 30. |
|
Point 3: Clin Exp Pharmacol Physiol. 2015 May;42(5):485-95. doi: 10.1111/1440-1681.12380. Can colonic migrating motor complexes occur in mice lacking the endothelin-3 gene? Kyra J Barnes 1, Nick J Spencer |
|
Response 3: Thank you so much for your advice. We have added this reference in our manuscript, which ranked 36. |

Reviewer 4 Report
Comments and Suggestions for Authors
Previous studies have investigated the temporal development of the ENS and onset of motility in mice. This study shows new information with regards to the PDGF2a cells and ICC and onset of colonic motility with RNA sequencing. The authors demonstrated the gradual maturation of ICC and PDGFRα+ cells, along with nitrergic and cholinergic nerves and purinergic receptors. The authors showed that in the CMC experiments, the proximal colon did not show detectable propulsive CMCs, and LNNA had no effect on the frequencies and amplitudes at PW1.There are some new findings here. However, many times work is described that has been done before, but not cited. The authors are not the first to show blocking NO synthesis increases CMCs. There are about 3-4 new references that need to be inserted and revised sentences to make this work up to date with the field.
The figures seem ok and English language ok.
Throughout the entire manuscript the word colonic migrating motor complex (CMMC) is used. This is an outdated term. Since a consensus terminology review was published in 2018 by Maura Corsetti and colleagues, the CMMC is abandoned and simply referred to as CMC (colonic motor complex). This is because it was agreed that not contractions (CMCs) migrate. Please change CMMC to CMC throughout the whole manuscript and change “colonic migrating motor complexe” to “colonic motor complex”, and quote this reference https://pubmed.ncbi.nlm.nih.gov/31296967/
It is unclear why the work of Roberts et al. (2007) was not cited ? See: https://pubmed.ncbi.nlm.nih.gov/17158255/. This paper is highly relevant to the current study under review. It needs to be cited and carefully discussed in relation to the data show here. Also this paper by Hao et al. (2010) is relevant and needs citation: https://pubmed.ncbi.nlm.nih.gov/20082666/
The introduction assumes the reader will know what a CMC is. There is no definition of CMCs. Hence, a statement needs to be included in the introduction. I suggest:
“In the adult mouse colon, a rhythmic neurogenic motor pattern is reliably recorded from the isolated whole mouse colon, called the colonic motor complex (CMC). A recent review describes the mechanisms known to underlie CMC generation in mouse (insert: https://pubmed.ncbi.nlm.nih.gov/33085903/) here. Also, many groups have studied CMCs and at least some should be cited.
Line 465 is needs some revision. The statement implies ICC generate CMCs. They don’t. CMCs are neurally-mediated, but may involve ICC. The mechanism underlying CMC generation has been shown to involved coordinated firing of the ENS. This needs to be quoted. I suggest replace the sentence “CMCs could by initiated by excitatory neurons through the pacemaker activity of ICCs” with “CMC generation has been shown to be due to coordinated firing of large populations of excitatory and inhibitory neurons at ~2Hz (insert: https://pubmed.ncbi.nlm.nih.gov/29807910/), but may involve ICC for neurotransmission.
Abstract: “rhythum” is spelt incorrectly.
Line 24: Replace this sentence which does not read well: “In PW5 mice, the regular pattern of CMMC was established with rhythum, similar to the typical murine adult CMMC pattern.” I suggest this:
“In PW5 mice, rhythmic CMCs were recorded, similar to the CMC pattern described previously in adult mice”.
Abstract: in this sentence “..increased with age grew from..” delete “grew”
Line 80: “…before puberty.” Is an inappropriate term. Suggest deleting this.
Line 444: Nitric oxide has been shown to be key for CMC generation. At the end of this sentence appropriate citation is needed: “NO is also involved in the inhibitory regulation of smooth muscle contraction and the 444 generation of CMMCs through ICCs [42, 43].” Also insert reference to these 3 laboratories: https://pubmed.ncbi.nlm.nih.gov/9198085/
https://pubmed.ncbi.nlm.nih.gov/31002480/
https://pubmed.ncbi.nlm.nih.gov/17158255/
Comments on the Quality of English Language
English is ok. Just lack of appropriate citations to relevant work in the past.
Author Response
Thank you very much for taking the time to review this manuscript. We have made modifications and supplements to the article according to your feedback, and have marked the modified areas in red, hoping to meet your requirements.
|
Point-by-point response to Comments and Suggestions for Authors |
|
Comments 1: Throughout the entire manuscript the word colonic migrating motor complex (CMMC) is used. This is an outdated term. Since a consensus terminology review was published in 2018 by Maura Corsetti and colleagues, the CMMC is abandoned and simply referred to as CMC (colonic motor complex). This is because it was agreed that not contractions (CMCs) migrate. Please change CMMC to CMC throughout the whole manuscript and change “colonic migrating motor complexe” to “colonic motor complex”, and quote this reference https://pubmed.ncbi.nlm.nih.gov/31296967/
|
|
Response 1: Thank you so much for your advice. We have changed all CMMC to CMC in the whole manuscript and quoted the reference.
|
|
Comments 2: It is unclear why the work of Roberts et al. (2007) was not cited ? See: https://pubmed.ncbi.nlm.nih.gov/17158255/. This paper is highly relevant to the current study under review. It needs to be cited and carefully discussed in relation to the data show here. Also this paper by Hao et al. (2010) is relevant and needs citation: https://pubmed.ncbi.nlm.nih.gov/20082666/ |
|
Response 2: Thank you so much for your advice. We have cited both references in the last paragraph of the introduction section and provided explanations. Our research objectives are different from those of these two laboratories. Hao mainly focused on the functional changes of the colon in embryonic mice, while Roberts mainly focused on the differences between CMMCs in P0, P2, P4, P6, P10, and adult mice. However, both studies only elucidate the developmental changes of ENS from a functional perspective. It also skipped the age range between P10 young mice and adult mice, which means there is a lack of juvenile stage. Our study supplemented the developmental changes of the colon in the age range of P10 to adult mice, and we simultaneously demonstrated the development of ENS and SIP syncytium in this age group from both molecular biology and functional perspectives. |
|
Comments 3: The introduction assumes the reader will know what a CMC is. There is no definition of CMCs. Hence, a statement needs to be included in the introduction. I suggest: “In the adult mouse colon, a rhythmic neurogenic motor pattern is reliably recorded from the isolated whole mouse colon, called the colonic motor complex (CMC). A recent review describes the mechanisms known to underlie CMC generation in mouse (insert: https://pubmed.ncbi.nlm.nih.gov/33085903/) here. Also, many groups have studied CMCs and at least some should be cited. |
|
Response 3: Thank you so much for your advice. We have added a statement of CMC in the introduction section and quoted relevant references. |
|
Comments 4: Line 465 is needs some revision. The statement implies ICC generate CMCs. They don’t. CMCs are neurally-mediated, but may involve ICC. The mechanism underlying CMC generation has been shown to involved coordinated firing of the ENS. This needs to be quoted. I suggest replace the sentence “CMCs could by initiated by excitatory neurons through the pacemaker activity of ICCs” with “CMC generation has been shown to be due to coordinated firing of large populations of excitatory and inhibitory neurons at ~2Hz (insert: https://pubmed.ncbi.nlm.nih.gov/29807910/), but may involve ICC for neurotransmission. |
|
Response 4: Thank you so much for your advice. We have changed this sentence with your suggestion and quoted this reference. |
|
Comments 5: Abstract: “rhythum” is spelt incorrectly. |
|
Response 5: Thank you so much for your advice. We have revised this mistake. |
|
Comments 6: Line 24: Replace this sentence which does not read well: “In PW5 mice, the regular pattern of CMMC was established with rhythum, similar to the typical murine adult CMMC pattern.” I suggest this: “In PW5 mice, rhythmic CMCs were recorded, similar to the CMC pattern described previously in adult mice”. |
|
Response 6: Thank you so much for your advice. We have changed this sentence with your suggestion. |
|
Comments 7: Abstract: in this sentence “..increased with age grew from..” delete “grew” |
|
Response 7: Thank you so much for your advice. We have changed this sentence with your suggestion. |
|
Comments 8: Line 80: “…before puberty.” Is an inappropriate term. Suggest deleting this. |
|
Response 8: Thank you so much for your advice. We have changed this sentence with your suggestion. |
|
Comments 9: Line 444: Nitric oxide has been shown to be key for CMC generation. At the end of this sentence appropriate citation is needed: “NO is also involved in the inhibitory regulation of smooth muscle contraction and the 444 generation of CMMCs through ICCs [42, 43].” Also insert reference to these 3 laboratories: https://pubmed.ncbi.nlm.nih.gov/9198085/ https://pubmed.ncbi.nlm.nih.gov/31002480/ https://pubmed.ncbi.nlm.nih.gov/17158255/ |
|
Response 9: Thank you so much for your advice. We have added articles from these three laboratories to the references in this sentence. |

Round 2
Reviewer 2 Report
Comments and Suggestions for Authors
The authors have fully responded to my comments.
However, the number of the subsection "Responses of CMCs with the application of MRS2500" is not "3.5.2", but "3.5.3".
Author Response
Thank you so much for your advice. We have revised this mistake.